# Hospital Pharmacists’ Attitudes and Intentions Toward Adverse Drug Reaction Reporting in Saudi Arabia: Insights from the Theory of Planned Behavior

**DOI:** 10.3390/healthcare13101111

**Published:** 2025-05-10

**Authors:** Fahad T. Alsulami

**Affiliations:** Clinical Pharmacy Department, College of Pharmacy, Taif University, Taif 21944, Saudi Arabia; f.alsulami@tu.edu.sa

**Keywords:** theory of planned behavior, adverse drug reaction, national pharmacovigilance center, hospital pharmacist, Saudi Arabia

## Abstract

**Objective:** This study assessed hospital pharmacists’ awareness, attitudes, and behaviors regarding adverse drug reaction (ADR) reporting and explored cognitive factors shaping hospital pharmacists’ intentions to report ADRs, using the theory of planned behavior (TPB) as a framework. **Methods:** A cross-sectional survey was conducted among hospital pharmacists from various regions of Saudi Arabia. Data were collected on their intentions to report ADRs to the national pharmacovigilance center (NPC), along with their attitudes, subjective norms, and perceived behavioral control related to ADR reporting. Descriptive statistics summarized the data, while multivariate logistic regression analyzed the influence of TPB constructs on reporting intentions. **Results:** A total of 141 hospital pharmacists participated in the study. While 86.5% of them were aware of the Saudi NPC, only 30% had reported ADRs in the past year. A strong intention to report ADRs was observed in 56% of the hospital pharmacists. Additionally, 53% exhibited a highly positive attitude, 57% perceived strong social norms, 52.5% reported high perceived behavioral control, and 63.8% expressed a strong moral obligation to report ADRs. Subjective norms and moral obligation emerged as significant predictors of the hospital pharmacists’ intention to report ADRs, according to the TPB constructs. **Conclusions:** While awareness of the Saudi NPC among hospital pharmacists was high, ADR reporting rates were low. Enhancing education, addressing barriers, and leveraging moral and normative drivers may strengthen pharmacovigilance practices and improve ADR reporting adherence among hospital pharmacists, ultimately fostering safer healthcare delivery.

## 1. Introduction

Adverse drug reactions (ADRs) are defined by the World Health Organization (WHO) as harmful and unintended responses to medications administered at standard dosages for the purposes of treatment, diagnosis, or physiological modification [1]. ADRs constitute a substantial burden on healthcare systems worldwide, contributing significantly to patient morbidity, prolonged hospitalization, increased healthcare costs, and, in severe cases, mortality [2,3,4]. Globally, ADRs are estimated to rank between the fourth and sixth leading causes of death, with financial repercussions exceeding USD 30.1 billion in the United States and EUR 79 billion across Europe annually [5]. In Saudi Arabia, institutional data have underscored the clinical burden of ADRs, with studies reporting notable rates of ADR-related hospital admissions, emergency department visits, and associated mortality across various healthcare settings [6,7,8].

The detection, documentation, and evaluation of ADRs are critical elements of pharmacovigilance, a scientific discipline dedicated to ensuring the safety and efficacy of pharmaceutical products through post-marketing surveillance [9]. However, underreporting of ADRs remains a globally recognized limitation of pharmacovigilance systems. Current estimates suggest that only 5% to 10% of ADRs are formally reported, undermining the capacity to identify emerging drug safety signals and implement corrective actions [10,11].

In Saudi Arabia, the Saudi Food and Drug Authority (SFDA) established the National Pharmacovigilance Center (NPC) in 2009, with the objective of streamlining ADR reporting mechanisms and encouraging healthcare professionals to contribute to national safety surveillance databases [12]. Despite the availability of electronic and paper-based reporting platforms, ADR reporting rates among healthcare providers in Saudi Arabia remain markedly low. Although comprising approximately 6% of the global population, Saudi Arabia and other Middle Eastern countries account for less than 0.6% of global pharmacovigilance reports [13,14].

Pharmacists, by virtue of their pharmacotherapeutic expertise and accessibility within clinical settings, are ideally positioned to play a pivotal role in ADR detection and reporting [15,16]. However, pharmacists have been consistently shown to underreport ADRs when compared to other healthcare professionals [17]. In Saudi Arabia, while many hospital pharmacists report encountering ADRs during their practice, the majority do not document these events through the NPC [18,19]. The literature attributes this underreporting to several barriers, including insufficient awareness of the NPC, lack of formal training, limited accessibility to reporting systems, and the absence of institutional incentives or dedicated time for completing reports [19,20,21,22].

To better understand the behavioral mechanisms underpinning ADR reporting practices, the theory of planned behavior (TPB) offers a comprehensive and empirically supported framework. The TPB posits that behavioral intention—defined as the motivational factor influencing a behavior—is determined by three constructs: attitude (the individual’s positive or negative evaluation of the behavior), subjective norms (perceived social pressure to perform or abstain from the behavior), and perceived behavioral control (the perceived ease or difficulty of performing the behavior given internal and external constraints) [23]. Previous applications of the TPB in pharmacovigilance research have demonstrated its explanatory power. For instance, Gavaza et al. (2011) reported that the TPB accounted for 34% of the variance in pharmacists’ intentions to report ADRs in the United States [24], while similar findings were observed in Malaysia, where TPB constructs explained 28% of the variance in reporting intentions among community pharmacists [25].

Despite the relevance of the TPB in predicting pharmacovigilance behaviors, no studies to date have applied this model to investigate hospital pharmacists’ ADR reporting practices in Saudi Arabia. The present study addressed this gap by employing the TPB as a theoretical lens to systematically examine the cognitive, attitudinal, and contextual factors influencing hospital pharmacists’ intentions to report ADRs at a national level. By assessing awareness of the NPC, attitudes toward ADR reporting, and perceived behavioral control, this research aimed to identify the key determinants that predict reporting intentions. The findings will contribute novel empirical insights and inform the design of targeted, evidence-based interventions aimed at enhancing pharmacovigilance engagement among hospital pharmacists in Saudi Arabia.

## 2. Materials and Methods

### 2.1. Participants

According to the Statistical Yearbook published by the Saudi Ministry of Health (MOH), as of 2023, there were 36,810 licensed pharmacists in Saudi Arabia, with approximately 30–35% employed in hospital settings [26]. In terms of gender distribution, the pharmacy workforce remains male-dominated. However, recent data indicate an increasing representation of female pharmacists, particularly in the western and central regions of the country [26].

This cross-sectional, web-based survey study employed a non-probability snowball sampling approach to target hospital pharmacists in Saudi Arabia. A self-administered survey was developed and distributed via Google Forms. Recruitment was facilitated through popular social media platforms, including WhatsApp and Telegram. To ensure a wide reach, the survey link was shared within specialized WhatsApp and Telegram groups that are specifically dedicated to hospital pharmacists. Additionally, individuals who identified themselves as hospital pharmacists on other platforms (e.g., X, formerly Twitter) were approached to participate in the survey and to further disseminate the survey link within their professional networks. College students contributed to the distribution process by sharing the survey link among their personal and professional contacts.

This study targeted hospital pharmacists across all regions of Saudi Arabia, including the central, eastern, western, northern, and southern regions. Efforts were made to ensure broad representation from these regions by actively recruiting participants from regional WhatsApp and Telegram groups and through targeted outreach. While the survey was anonymous and no identifiable personal information was collected, demographic questions included location data to validate the geographic distribution of the respondents. The survey included an informed consent statement that outlined the study objectives, ensured participant confidentiality, and confirmed voluntary participation. Pharmacists were eligible for inclusion in this study if they were licensed professionals registered with the Saudi Commission for Health Specialties (SCFHS) and were currently practicing within a hospital pharmacy setting in Saudi Arabia. Exclusion criteria included pharmacists who lacked SCFHS licensure, those who were pharmacy interns or undergraduate students, and individuals employed in non-hospital pharmacy settings.

### 2.2. Study Instruments

A structured questionnaire, adapted from previously validated instruments, was employed in this study [24,25]. The survey encompassed sections addressing sociodemographic characteristics, awareness of the Saudi NPC and the ADR reporting system, hospital pharmacists’ ADR reporting behaviors, and constructs derived from the TPB. Additionally, a measure of perceived moral obligation was included. The survey instruments were informed by a review of the literature and underwent evaluation by two academic experts. Based on their feedback, necessary modifications were implemented, including replacing the term “FDA” with “Saudi NPC” to ensure the terminology aligned with the local context and regulatory framework. The validity of the questionnaire was assessed through face and content validity checks via a pilot study involving 10 pharmacists, whose responses were not included in the final analysis. To ensure clarity and minimize any linguistic ambiguity, the survey was available in both English and Arabic. Furthermore, the reliability of the TPB constructs was evaluated using Cronbach’s alpha to assess internal consistency.

#### 2.2.1. TPB Constructs

The TPB framework was operationalized through the measurement of four key constructs: intention, attitude, subjective norms, and perceived behavioral control. All items used to assess these constructs were adapted from previously validated scales [24,25]. Intention to report ADRs was measured through three items (e.g., “I intend to report serious ADRs to the Saudi NPC”), with responses captured on a 7-point Likert scale ranging from “extremely disagree” to “extremely agree”. Scores ranged from 1 to 7, where higher scores indicated stronger intentions to report ADRs. The median score was used to dichotomize participants into low-intention (coded as 0) and high-intention (coded as 1) groups.

Attitude toward ADR reporting was assessed using five items (e.g., “Reporting ADRs to the NPC is valuable”), also rated on a 7-point Likert scale. Higher scores indicated more favorable attitudes. Participants were similarly categorized into low- (coded as 0) and high- (coded as 1) attitude groups based on the median score.

Subjective norms were measured by three items (e.g., “Most people important to me think I should report ADRs”), again on a 7-point Likert scale. A higher score indicated greater perceived social pressure to report ADRs. Based on the median score, participants were classified into low subjective norms (coded as 0) and high subjective norms (coded as 1).

Perceived behavioral control was assessed through two items (e.g., “I believe I have control over reporting ADRs”), using the same 7-point Likert scale. Higher scores indicated stronger perceived control over ADR reporting. Participants were grouped into low control (coded as 0) and high control (coded as 1) based on the median score.

#### 2.2.2. Perceived Moral Obligation

Perceived moral obligation was measured using a single item (e.g., “I feel morally obligated to report ADRs to the NPC”), with responses recorded on a 7-point Likert scale. A higher score indicated a stronger sense of moral responsibility to report ADRs. Participants were classified into low moral obligation (coded as 0) and high moral obligation (coded as 1) based on the median score.

### 2.3. Statistical Analysis

Descriptive statistics were employed to summarize the data, including frequencies, percentages, means, standard deviations (SDs), medians, and interquartile ranges (IQRs). To examine the associations between sociodemographic variables and pharmacists’ intention to report ADRs, chi-square tests were performed. The dependent variable in this analysis was pharmacists’ intention to report ADRs, which was categorized into two levels: “Hospital Pharmacist with high intention to report ADRs to Saudi NPC” and “Hospital Pharmacist with low intention to report ADRs to Saudi NPC”. Independent variables included various sociodemographic factors, such as age, gender, education, years of experience, region of work in Saudi Arabia, number of pharmacists working in the same shift, awareness of the existence of NPC in Saudi Arabia, familiarity with the ADR reporting process in Saudi Arabia, previous training on ADR reporting, and whether the pharmacist had reported any ADRs to the Saudi NPC in the past 12 months. Additionally, TPB-related factors such as attitude, subjective norms, perceived behavioral control, and perceived moral obligation were included as independent variables.

Additionally, an entry multivariable logistic regression analysis was performed to examine the influence of constructs from the theory of planned behavior (TPB) and perceived moral obligation on the intention to report ADRs. The dependent variable in this analysis was also pharmacists’ intention to report ADRs. Independent variables included TPB constructs (attitudes, subjective norms, perceived behavioral control) and perceived moral obligation. Statistical significance was determined using a *p*-value threshold of <0.05 and a 95% confidence interval (CI). All analyses were conducted using IBM^®^ SPSS Statistics, version 29.0.

## 3. Results

### 3.1. Respondent Profile

Table 1 presents the sociodemographic characteristics of the hospital pharmacists included in the study, with a total sample size of 141 participants. The mean age of the pharmacists was 31.02 years (SD = 5.61). The majority of participants were male (61%), with females representing 39% of the sample. Regarding educational qualifications, most pharmacists held a Doctor of Pharmacy (PharmD) degree (43.3%), followed by a bachelor’s degree (29.1%), a graduate education degree (17.7%), and a diploma (9.9%). The pharmacists predominantly worked in the western region of Saudi Arabia (46.8%), with fewer participants from the central (27.7%), southern (14.9%), eastern (7.8%), and northern (2.8%) regions of Saudi Arabia.

Experience in hospital pharmacy settings varied, with nearly half of pharmacists having fewer than 5 years of experience (46.8%), 32.6% having 5 to 10 years, and 20.6% having more than 10 years. A significant majority were aware of the Saudi NPC (86.5%) and familiar with the ADR reporting process (75.2%). Despite this, only 29.8% had reported any ADRs in the past 12 months, highlighting a gap between awareness and actual reporting behavior.

### 3.2. Intention to Report ADRs

The intention construct demonstrated strong internal consistency, with a Cronbach’s alpha value of 0.88. A significant majority of hospital pharmacists (87.9%) agreed that they intend to report serious ADRs. Similarly, 87.9% indicated they would try to report serious ADRs. When asked if they plan to report serious ADRs, 85.1% agreed. The findings show that hospital pharmacists had a strong intention to report serious ADRs they encounter to the Saudi NPC, with an overall mean score of 6.20 (SD = 1.20) out of 7. The median score was 7 (IQR = 1.33). As a result, about 56% of hospital pharmacists had a high intention to report ADRs to the Saudi NPC (Table 2).

### 3.3. TPB Constructs and Perceived Moral Obligation

The attitude construct showed reliable internal consistency, with a Cronbach’s alpha value of 0.73. A substantial 89.4% of the hospital pharmacists agreed that reporting ADRs is valuable. The perception that reporting ADRs is pleasant was slightly lower, with 85.1% agreeing and 9.9% disagreeing. The statement that reporting ADRs is beneficial received the highest agreement, with 93.6% of pharmacists affirming it. However, there was more variability in the belief that reporting ADRs is enjoyable, where only 67.4% agreed and 14.2% disagreed. The pharmacists’ attitudes toward reporting ADRs were highly positive, with a mean score of 6.28 (SD = 1.01) out of 7. The median was 6.60 (IQR = 1.20). Consequently, about 53% of hospital pharmacist perceived a high positive attitude (Table 2).

The subjective norm construct exhibited reliable internal consistency, with Cronbach’s alpha producing a value of 0.77. A majority of hospital pharmacists (61.7%) agreed that people important to them think they should report ADRs. The belief that valued individuals in their lives would approve of their ADR reporting was supported by 80.9% of respondents, with a minimal 4.3% disagreement. Additionally, 71.6% indicated that the pharmacists whose opinions they value also report ADRs, although 8.5% disagreed. The subjective norms surrounding ADR reporting among hospital pharmacists had a mean score of 5.57 (SD = 1.32) out of 7. The median score was 5.66 (IQR = 2.33) indicating that about 57% of hospital pharmacists perceived a high social norms form peers and significant others (Table 2).

About 64.5% of hospital pharmacists agreed that they have complete control over reporting ADRs. Additionally, only 52.5% felt that it is mostly up to them whether or not they report ADRs, with a notable 21.3% disagreeing. Perceived behavioral control had a lower mean score of 4.86 (SD = 1.48) out of 7. The median score was 5 (IQR = 2), resulting in 52.5% of hospital pharmacists who perceived they had high control in reporting ADRs to the Saudi NPC (Table 2).

A significant 85.1% of hospital pharmacists agreed that they have a moral obligation to report ADRs they encounter, while only 3.5% disagreed. This highlights that a strong ethical component drives pharmacists’ intention to report ADRs, reinforcing the importance of moral responsibility in pharmacovigilance practices. The sense of moral obligation to report ADRs was strong among the pharmacists, with a mean score of 6.18 (SD = 1.34) out of 7. The median score was 7 (IQR = 1). Consequently, 63.8% of hospital pharmacists perceived a high moral obligation to report ADRs to the Saudi NPC (Table 2).

### 3.4. ADR Reporting Intentions Among Hospital Pharmacists by Category

The results presented in Table 3 illustrate the distribution of hospital pharmacists with a high intention to report ADRs to the Saudi NPC across various sociodemographic and professional categories. In terms of gender, a greater proportion of female hospital pharmacists (65.5%) demonstrated a high intention to report ADRs compared to their male counterparts (61.6%), although this difference did not reach statistical significance (*p* = 0.071). Regarding educational attainment, hospital pharmacists holding a bachelor’s degree exhibited the highest proportion of high intention (68.3%), followed by those with a diploma (57.1%), Doctor of Pharmacy (PharmD) degrees (50.8%), and graduate degrees (48%). Nevertheless, the association between educational level and intention to report ADRs was not statistically significant (*p* = 0.280). Regional variation was also observed; notably, all pharmacists from the northern region reported high intention (100%), whereas those in the central region had the lowest intention (51.3%). However, these regional differences were not statistically significant (*p* = 0.261).

Similarly, neither the number of years of professional experience in a hospital setting nor the number of pharmacists working per shift showed a statistically significant relationship with reporting intention (*p* = 0.931 and *p* = 0.581, respectively). Additionally, prior experience in reporting ADRs was not significantly associated with current intention to report (*p* = 0.097). Awareness of the NPC and familiarity with ADR reporting procedures were also not significantly correlated with reporting intention (*p* = 0.242 and *p* = 0.070, respectively).

In contrast, participation in ADR reporting training was significantly associated with increased intention (*p* = 0.023), with 62.8% of trained hospital pharmacists expressing high intention compared to 42.6% of untrained hospital pharmacists. Furthermore, attitude toward ADR reporting was significantly associated with intention (*p* = 0.007); hospital pharmacists exhibiting a highly positive attitude were more likely to report a high intention (66.7%) than those with a less positive attitude (43.9%). Subjective norms also emerged as a strong predictor of reporting intention (*p* < 0.001). Hospital pharmacists perceiving strong social pressure to report ADRs demonstrated markedly higher intention (77.5%) than those perceiving weaker social pressure (27.9%). Similarly, perceived moral obligation was a significant predictor (*p* < 0.001), with those reporting a high moral duty to report ADRs showing greater intention (76.7%) than those with a lower sense of moral responsibility (19.6%). However, perceived behavioral control was not significantly associated with intention to report ADRs (*p* = 0.123).

### 3.5. Predictors Associated with the Intention of Hospital Pharmacists to Report ADRs

A binary logistic regression model was conducted to estimate the effect of predictors, including TPB constructs and the perceived moral obligation construct, on the dependent variable (intention to report ADRs to the Saudi NPC). Overall, the model was significant (chi-square = 61.60, *p* < 0.001) and explained 47% of the variance in reporting intention based on the Nagelkerke *R*^2^ value. The results from the binary logistic regression model demonstrated significant findings for perceived subjective norms and perceived moral obligation.

Hospital pharmacists who had high perceptions of subjective norms were found to be significantly more likely to report ADRs, with an odds ratio (OR) of 7.42 (95% CI: 2.54–21.69, *p* < 0.001). This suggests that social pressures or expectations from others played a crucial role in shaping their reporting intentions. Similarly, perceived moral obligation had a strong positive association with reporting intention, yielding an OR of 7.83 (95% CI: 3.09–19.86, *p* < 0.001). Hospital pharmacists who felt a higher moral duty were far more likely to engage in reporting, indicating that ethical considerations heavily influence their behavior (Table 4).

Conversely, attitude toward reporting and perceived behavioral control did not show a statistically significant impact. High positive attitudes (OR = 1.57, *p* = 0.327) and high perceived behavioral control (OR = 0.38, *p* = 0.090) were not strong predictors of reporting intentions. This suggests that while pharmacists may feel capable or view reporting positively, these factors alone do not significantly drive the intention to report ADRs among hospital pharmacists.

## 4. Discussion

ADRs continue to be a leading cause of morbidity and mortality globally, ranking among the primary contributors to death and illness [27,28]. Timely and accurate ADR reporting is essential for the ongoing surveillance of drug safety and effectiveness. Nevertheless, it is estimated that only 5% to 10% of ADRs are reported worldwide, indicating a significant gap in reporting [11]. As medication experts, pharmacists are uniquely positioned to identify and report ADRs, thereby playing a pivotal role in minimizing the risks associated with drug therapies [15]. This study aimed to evaluate the awareness, attitudes, and practices of hospital pharmacists concerning ADR reporting. Additionally, it sought to identify cognitive determinants influencing their intention to report ADRs, utilizing the framework of the TPB.

A substantial proportion of hospital pharmacists demonstrated awareness of the Saudi NPC, with 86.5% being familiar with its existence and 75.2% being knowledgeable about the ADR reporting process. This high level of awareness and familiarity aligns closely with findings from studies conducted in other regions of Saudi Arabia, which have consistently reported strong awareness and engagement with pharmacovigilance systems among healthcare professionals [19,29]. These results highlight the effectiveness of ongoing educational initiatives and the integration of pharmacovigilance into professional practice among healthcare professionals in the country.

However, despite this high awareness, this study revealed that only 30% of hospital pharmacists had reported any ADRs in the past 12 months, indicating a significant gap between awareness and actual reporting practices. This discrepancy reflects a common challenge in pharmacovigilance systems, where knowledge and familiarity do not always translate into proactive reporting behavior. Similar patterns have been observed in studies conducted both in Saudi Arabia and across other Gulf countries, where low ADR reporting rates persist despite high levels of awareness [2,30,31]. Similarly, low rates of ADR reporting have been observed among hospital pharmacists in various regions worldwide [32,33]. This suggests the need for targeted interventions to address barriers to ADR reporting, such as time constraints, work load, or lack of confidence in identifying ADRs [34,35]. Enhancing the reporting culture through supportive policies, training workshops, and simplifying the reporting process could help bridge this gap and improve pharmacovigilance outcomes.

Over half of hospital pharmacists in this study demonstrated a high intention to report ADRs to the Saudi NPC, which aligns with findings from previous research conducted in various regions of Saudi Arabia. For instance, Alshabi et al. (2022) reported similar levels of intention among pharmacists in the Najran region, suggesting a nationwide consistency in the willingness to report ADRs to the Saudi NPC. These findings are further corroborated by Abdulsalim et al. (2023), who observed comparable trends in pharmacists’ willingness to engage in ADR reporting behaviors in the Al-Qassim region. Collectively, these studies highlight the critical role of pharmacists as frontline reporters of ADRs, contributing significantly to a robust pharmacovigilance system across different regions of Saudi Arabia [19,36]. Moreover, the alignment of findings across regions reinforces the significance of fostering a unified national strategy to enhance ADR reporting practices. The results indicate that a higher proportion of female hospital pharmacists (65.5%) reported a high intention to report ADRs compared to their male counterparts (50%), though this difference was not statistically significant (*p* = 0.071). This trend could potentially be explained by the possibility that female pharmacists possess greater knowledge or awareness of pharmacovigilance practices in Saudi Arabia, which may enhance their motivation to report ADRs [36]. However, as the difference was not statistically significant, it is crucial to interpret this cautiously. Further research is necessary to investigate the underlying factors, such as differences in training, exposure, or attitudes toward pharmacovigilance, that might explain the observed trend.

This study revealed that approximately 53% of hospital pharmacists demonstrated a strong positive attitude toward reporting ADRs. However, this percentage is lower than the findings of similar studies conducted in specific regions, such as Najran, Majmaah, and Alhasa, where higher levels of positive attitudes toward ADR reporting were observed [19,29,37]. This discrepancy may be attributed to regional differences in awareness, training, or institutional policies regarding ADR reporting. Additionally, variations in sample size or study methodology could also contribute to these differences. These findings highlight the need for targeted interventions to improve the attitudes and practices of hospital pharmacists regarding ADR reporting on a broader national scale. Strengthening training programs, increasing awareness campaigns, and implementing uniform policies across all regions could bridge the gap and enhance the overall effectiveness of pharmacovigilance efforts in Saudi Arabia.

In this study, approximately 57% of hospital pharmacists reported perceiving strong subjective norms from their peers and other significant individuals regarding the reporting of ADRs. This finding aligns with the results of several previous studies [24,25]. This suggests that a majority of pharmacists feel social pressure or influence from colleagues and other stakeholders to actively engage in ADR reporting. The perception of high subjective norms may play a crucial role in motivating pharmacists to report ADRs, as individuals are often more likely to take actions that align with the expectations of those around them.

A little over half (52.5%) of hospital pharmacists reported having high behavioral control when it comes to reporting ADRs to the Saudi NPC. This figure suggests a moderate level of confidence in their ability to report ADRs. However, this percentage is relatively low when compared to findings from similar studies conducted in the United States and Malaysia, where a significantly higher proportion of pharmacists reported feeling high control over ADR reporting (81%) in the United States and (74%) in Malaysia may reflect differences in the regulatory environments, training, and resources available to pharmacists in these countries [24,25]. In Saudi Arabia, the reporting of ADRs may be influenced by factors such as lack of knowledge about ADRs, insufficient training, or limitations in the infrastructure for pharmacovigilance [19,38,39]. Moreover, cultural and institutional factors could also play a role in pharmacists’ perceptions of their ability to report ADRs effectively. The lower confidence observed in this study suggests that there may be a need for further efforts to enhance the support and education for hospital pharmacists in Saudi Arabia to increase their sense of control and competence in the ADR reporting process.

This study revealed that approximately 64% of hospital pharmacists in Saudi Arabia perceived a high moral obligation to report ADRs to the Saudi NPC. While this figure demonstrates a majority, it is relatively low compared to findings from similar studies conducted in other countries. For instance, studies in the United States and Malaysia reported significantly higher proportions of pharmacists expressing a moral obligation to report ADRs [24,25]. These discrepancies may reflect differences in training, cultural attitudes toward pharmacovigilance, and the integration of ADR reporting into professional practices across countries. Strengthening education and promoting awareness in Saudi Arabia could potentially improve hospital pharmacists’ commitment to ADR reporting.

The findings of this study demonstrate that the constructs of perceived subjective norms and perceived moral obligation were significantly correlated with the intention of hospital pharmacists to report ADRs to the Saudi NPC, within the framework of the TPB. This suggests that hospital pharmacists are more likely to exhibit an intention to report ADRs when they perceive those significant others (e.g., colleagues, supervisors, or professional organizations) expect and value such reporting behaviors, as captured by the subjective norm construct. Similarly, the perceived moral obligation reflects an intrinsic sense of duty or ethical commitment to ensure patient safety and improve pharmacovigilance practices, further driving the intention to report ADRs. These results align with previous research, which highlights that subjective norms often play a pivotal role in shaping healthcare professionals’ behavioral intentions in contexts where peer and organizational expectations are highly influential [24,25,40,41]. Furthermore, the strong association with perceived moral obligation underscores the critical role of ethical and moral considerations in motivating pharmacists to engage in ADR reporting, a finding consistent with studies in similar healthcare settings [24,25].

Building on the identified significance of subjective norms and moral obligation, this study emphasizes the critical role of fostering a collective sense of responsibility and ethical commitment among hospital pharmacists. Practical intervention strategies could include structured organizational training programs that highlight the value of collaborative pharmacovigilance, fostering peer support and mentorship networks to reinforce professional norms. Ethical reinforcement strategies, such as case-based discussions and recognition of exemplary reporting behaviors, could strengthen pharmacists’ moral obligation and accountability. These recommendations align with findings from previous studies in pharmacovigilance, such as [40,42], which demonstrate that interventions targeting professional norms and ethical values effectively enhance reporting behaviors. Moreover, such strategies could be tailored to address the unique cultural and professional dynamics in Saudi Arabia, ensuring both relevance and sustainability in strengthening ADR reporting systems.

## 5. Limitations

This study has several notable limitations. First, its cross-sectional design precludes the establishment of causal inferences. Second, the use of a self-reported survey may introduce recall bias, potentially leading to either overestimation or underestimation of the observed associations. Third, the adoption of a snowball sampling method may have resulted in selection bias by recruiting participants within similar professional or social circles, thereby limiting the diversity of perspectives and reducing the generalizability of the findings to the broader population of hospital pharmacists in Saudi Arabia. Fourth, this study focused exclusively on hospital pharmacists, underscoring the need for future research to encompass pharmacists from various practice settings. Fifth, the sample predominantly comprised individuals from the western region of Saudi Arabia, which may compromise the representativeness of the results at the national level. Sixth, the relatively small sample size may further constrain the generalizability of the findings and elevate the risk of statistical bias. Lastly, the absence of a standardized instrument to measure pharmacists’ intention to report ADRs within the framework of the TPB represents an additional limitation.

## 6. Conclusions

This study provided an in-depth evaluation of hospital pharmacists’ attitudes, intentions, and practices toward ADR reporting in Saudi Arabia, guided by the TPB. The findings revealed that while most pharmacists demonstrated awareness and familiarity with the ADR reporting process, only one-third actively participated in reporting ADRs to the Saudi NPC. Furthermore, two-thirds of pharmacists exhibited positive attitudes and a strong sense of moral obligation toward ADR reporting, but only half reported high levels of perceived subjective norms and behavioral control. Importantly, perceived subjective norms and moral obligation emerged as key predictors of pharmacists’ intentions to report ADRs. To enhance ADR reporting practices across Saudi Arabia, it is essential to foster a culture of peer support and professional accountability through awareness campaigns, mentoring, and workshops that emphasize the collective responsibility for ADR reporting. Additionally, integrating ethical education into pharmacovigilance training programs can strengthen pharmacists’ sense of moral obligation by highlighting the critical role of ADR reporting in patient safety and public health. These targeted interventions can significantly improve ADR reporting behaviors and strengthen the pharmacovigilance system in Saudi Arabia.

## Figures and Tables

**Table 1 healthcare-13-01111-t001:** Sociodemographic characteristics of the hospital pharmacists (n = 141).

Sociodemographic Variables	Frequency	Percent
**Age**	Mean: 31.02 years old	SD: 5.61
**Gender**		
Female	55	39%
Male	86	61%
**Educational Degree**		
Diploma degree in pharmacy	14	9.9%
Bachelor’s degree in pharmacy	41	29.1%
Doctor of Pharmacy (PharmD)	61	43.3%
Graduate education degree in pharmacy	25	17.7%
**Region of work in Saudi Arabia**		
Western region	66	46.8%
Eastern region	11	7.8%
Central region	39	27.7%
Northern region	4	2.8%
Southern region	21	14.9%
** Experience as pharmacist in hospital setting **		
Fewer than 5 years	66	46.8%
From 5 to 10 years	46	32.6%
More than 10 years	29	20.6%
** Number of pharmacists working in the same shift in the hospital pharmacy **		
Fewer than 5 pharmacists	63	44.7%
From 5 to 10 pharmacists	51	36.2%
More than 10 pharmacists	27	19.1%
** Awareness of the existence of NPC in Saudi Arabia **		
Not aware	19	13.5%
Aware	122	86.5%
**Familiar with the way of ADR reporting in Saudi Arabia**		
Not Familiar	35	24.8%
Familiar	106	75.2%
** Have you been trained to report ADRs before? **		
No	47	33.3%
Yes	94	66.7%
**Have you reported any ADRs to the Saudi NPC in in the previous 12 months?**		
No	99	70.2%
Yes	42	29.8%

**Table 2 healthcare-13-01111-t002:** Intention, attitude, subjective norm, perceived behavioral control, and perceived moral obligation of hospital pharmacists to report ADRs to the Saudi NPC.

Variables	Disagree	Neutral	Agree
**Intention**	Mean: 6.20, SD: 1.20 Median: 7, IQR: 1.33		
I intend to report serious ADRs that I will encounter to the Saudi NPC.	4.3%	7.8%	87.9%
I will try to report serious ADRs that I will encounter to the Saudi NPC.	4.3%	7.8%	87.9%
I plan to report serious ADRs that I will encounter to the Saudi NPC.	7.1%	7.8%	85.1%
**Attitudes**	Mean: 6.28, SD: 1.01 Median: 6.60, IQR: 1.20		
Reporting ADRs to NPC in Saudi Arabia is valuable.	6.4%	4.3%	89.4%
Reporting ADRs to NPC in Saudi Arabia is pleasant.	9.9%	5.0%	85.1%
Reporting ADRs to NPC in Saudi Arabia is good.	1.4%	2.1%	89.4%
Reporting ADRs to NPC in Saudi Arabia is enjoyable.	14.2%	18.4%	67.4%
Reporting ADRs to NPC in Saudi Arabia is beneficial.	3.5%	2.8%	93.6%
**Subjective Norms**	Mean: 5.57, SD: 1.32 Median: 5.66, IQR: 2.33		
Most people who are important to me think that I should report ADRs that I encounter to NPC in Saudi Arabia.	13.5%	24.8%	61.7%
The people in my life whose opinions I value would approve my reporting of ADRs that I encounter to NPC in Saudi Arabia.	4.3%	14.9%	80.9%
The pharmacists whose opinions I value report ADRs to NPC in Saudi Arabia.	8.5%	19.9%	71.6%
**Perceived Behavioral Control**	Mean: 4.86, SD: 1.48 Median: 5, IQR: 2		
You believe you have complete control over reporting ADRs that you encounter to NPC in Saudi Arabia.	14.9%	20.6%	64.5%
It is mostly up to me whether or not I report ADRs to NPC in Saudi Arabia.	21.3%	26.2%	52.5%
**Perceived Moral Obligation**	Mean: 6.18, SD: 1.34 Median: 7, IQR: 1		
I believe I have a moral obligation to report ADRs that I will encounter to NPC in Saudi Arabia.	3.5%	11.3%	85.1%

Note: participants who very strongly disagreed or disagreed (1 to 3) were grouped into the disagree category, and those who very strongly agreed or agreed (5 to 7) were grouped into the agree category.

**Table 3 healthcare-13-01111-t003:** Rate of hospital pharmacist with high intention to repot ADRs to the Saudi NPC across different categories.

Variables	Hospital Pharmacists with Low Intention to Report ADRs to the Saudi NPC (n = 62)	Hospital Pharmacists with High Intention to Report ADRs to the Saudi NPC (n = 79)	*p*-Value
**Gender**			0.071
Female	34.5%	65.5%	
Male	50.0%	50.0%	
** Educational Degree **			0.280
Diploma degree in pharmacy	42.9%	57.1%	
Bachelor’s degree in pharmacy	31.7%	68.3%	
Doctor of Pharmacy (PharmD)	49.2%	50.8%	
Graduate education degree in pharmacy	52.0%	48.0%	
**Region of work in Saudi Arabia**			0.261
Western region	48.5%	51.5%	
Eastern region	36.4%	63.6%	
Central region	48.7%	51.3%	
Northern region	0.0%	100.0%	
Southern region	33.3%	66.7%	
** Experience as pharmacist in hospital setting **			0.931
Fewer than 5 years	45.5%	54.5%	
From 5 to 10 years	43.5%	56.5%	
More than 10 years	41.4%	58.6%	
** Number of pharmacists working in the same shift in the hospital pharmacy **			0.581
Fewer than 5 pharmacists	42.9%	57.1%	
From 5 to 10 pharmacists	49.0%	51.0%	
More than 10 pharmacists	37.0%	63.0%	
** Awareness of the existence of NPC in Saudi Arabia **			0.242
Not aware	31.6%	68.4%	
Aware	45.9%	54.1%	
** Familiar with the way of ADR reporting ** **in Saudi Arabia**			0.070
Not Familiar	57.1%	42.9%	
Familiar	39.6%	60.4%	
** Have you been trained to report ADRs before? **			0.023
No	57.4%	42.6%	
Yes	37.2%	62.8%	
**Have you reported any ADRs to the Saudi NPC in in the previous 12 months?**			0.097
No	48.5%	51.5%	
Yes	33.3%	66.7%	
**Attitude**			0.007
Perceived low positive attitude	56.1%	43.9%	
Perceived high positive attitude	33.3%	66.7%	
**Subjective norms**			< 0.001
Perceived low subjective norms	72.1%	27.9%	
Perceived high subjective norms	22.5%	77.5%	
**Perceived behavioral control**			0.123
Perceived low behavioral control	50.7%	49.3%	
Perceived high behavioral control	37.8%	62.2%	
**Perceived moral obligation**			< 0.001
Perceived low moral obligation	80.4%	19.6%	
Perceived high moral obligation	23.3%	76.7%	

**Table 4 healthcare-13-01111-t004:** Multivariate logistic regression model of the intention to report ADRs to the Saudi NPC of hospital pharmacists.

Predictor Variables	Coefficient	SE	Wald	OR	Lower 95% CI	Upper 95% CI	*p*-Value
**Attitude**							
Perceived low positive attitude	Reference						
Perceived high positive attitude	0.45	0.46	0.96	1.57	0.63	3.89	0.327
**Subjective norms**							
Perceived low subjective norms	Reference						
Perceived high subjective norms	2.00	0.54	13.42	7.42	2.54	21.69	<0.001
**Perceived behavioral control**							
Perceived low behavioral control	Reference						
Perceived high behavioral control	−0.96	0.56	2.88	0.38	0.12	1.16	0.090
**Perceived moral obligation**							
Perceived low moral obligation	Reference						
Perceived high moral obligation	2.05	0.47	18.80	7.83	3.09	19.86	<0.001

## Data Availability

The data are available upon request.

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
