# Peer review of "Hospital Pharmacists’ Attitudes and Intentions Toward Adverse Drug Reaction Reporting in Saudi Arabia: Insights from the Theory of Planned Behavior"

_healthcare, 2025, doi:10.3390/healthcare13101111_

Round 1
Reviewer 1 Report
Comments and Suggestions for Authors
Due to the lack of novelty and the use of outdated and merely limited to low-quality references, I must reject the manuscript.
Reviewer 2 Report
Comments and Suggestions for Authors
I express my gratitude for the invitation. The authors surveyed to investigate the attitudes and intentions of hospital pharmacists concerning adverse drug reactions (ADRs). The results indicated that while 86.5% of the respondents were aware of the 16 guidelines established by the Saudi National Pharmacovigilance Center (NPC), only 30% had reported any ADRs in the past year. This study highlights a significant issue, as the rate of ADR reporting among pharmacists is markedly low. The findings are undoubtedly significant; however, the authors should emphasize the study's novelty. the introduction section should be shortened.
Reviewer 3 Report
Comments and Suggestions for Authors
- The introduction is too lengthy. Try to limit it to just one page.
- It appears that most of the study participants were not directly approached by the survey team. Instead, participation was achieved through a Google form link that was shared on social media platforms. So how did the investigators ensure or verify that their participants were indeed pharmacists? Although the link was initially shared within “specialized or professional chat groups,” what was done to prevent the link from being forwarded by members of the professional group to other non-professional groups to prevent the participation of non-pharmacists?
- Line 129: it was stated that pharmacists were included in the study based on the following criteria: being registered professionals in Saudi Arabia and currently employed in a hospital pharmacy setting. How was this determined since the “survey was anonymous and no identifiable personal information was collected?” Did the investigators rely 100% on the participants declaration alone and did nothing to verify their claims?
- In Table 2, many of the variables listed are subjective. For intention, what is the difference between “plan to” and “intend?” For attitudes, how would one differentiate “good” from “pleasant,” “enjoyable” or even “beneficial?” All of these terms have a positive meaning and are also synonyms of the word “good.”
- As a point for discussion, I wonder why pharmacists with a higher level degree (PharmD) appear to be less inclined (compared to those with just a bachelor’s degree) to report ADRs?
- Merely stating how many pharmacists work in the same shift does not necessarily reflect the actual workload. For example, a general hospital pharmacy with more than 10 pharmacists per shift may still be very busy compared to a district hospital pharmacy with 5-10 pharmacists per shift. Thus a better indicator would be the pharmacist to patient ratio.
- Also, could a “carrot and stick” system be introduced to encourage the reporting of ADRs?
Reviewer 4 Report
Comments and Suggestions for Authors
Thank you for the opportunity to review this paper. It is an interesting paper on an important topic and I enjoyed reading it. I have the following comments/suggestions:
Introduction:
- It would be useful to provide more context for statistics cited. For example, you note that Saudi Arabia, along with other Middle Eastern nations, contributes to only 0.6% of global safety reports. How does this compare to their population (e.g. if this region accounts for only 0.6% of the global population this is OK but if the population is greater than would indicate underreporting).
- While the introduction provides a good overview of existing work and contextualises your study, some parts could be expressed more succinctly. For example, rather than describing each Saudi Arabian study examining ADR reporting in detail, this information could be synthesised into a single sentence.
Methods:
- Please clarify what being a registered professional means- how is this different to just being a pharmacist working in a hospital (is there a difference)?
Results:
- It would be useful to provide some context for the demographics of your participants. For example, how does this compare to Saudi Arabian pharmacists generally, hospital pharmacists and hospital pharmacists in the Western region?
- Was it possible to calculate a response rate? If not, can you provide more context for your sample size (e.g. by giving an estimate of the number of hospital pharmacists working in the Western region of Saudi Arabia who were eligible to participate?
Discussion:
- Your discussion nicely contextualised the problem you were trying to address and your results, with sound policy suggestions. However, is there any value comparing your results to studies from other parts of the world (rather than predominantly the Middle East)?
I look forward to reading the next version of this paper!
Reviewer 5 Report
Comments and Suggestions for Authors
See comments with the file attached.

Round 2
Reviewer 1 Report
Comments and Suggestions for Authors
The author has been fulfilled the comments arrived and response to the all queries reasonably, hence I recommend the publication of the manuscript in the Healthcare journal as is.
Reviewer 3 Report
Comments and Suggestions for Authors
The authors have managed to address all of my concerns adequately. I have no further comments or queries.
Reviewer 5 Report
Comments and Suggestions for Authors
Thank you for thoughtful consideration of comments.